# MACHINE TRANSLATION WITH WEAKLY PAIRED BILINGUAL DOCUMENTS

## ABSTRACT

Neural machine translation, which achieves near human-level performance in some languages, strongly relies on the availability of large amounts of parallel sentences, which hinders its applicability to low-resource language pairs. Recent works explore the possibility of unsupervised machine translation with monolingual data only, leading to much lower accuracy compared with the supervised one. Observing that weakly paired bilingual documents are much easier to collect than bilingual sentences, e.g., from Wikipedia, news websites or books, in this paper, we investigate the training of translation models with weakly paired bilingual documents. Our approach contains two components/steps. First, we provide a simple approach to mine implicitly bilingual sentence pairs from document pairs which can then be used as supervised signals for training. Second, we leverage the topic consistency of two weakly paired documents and learn the sentence-to-sentence translation by constraining the word distribution-level alignments. We evaluate our proposed method on weakly paired documents from Wikipedia on four tasks, the widely used WMT16 German↔English and WMT13 Spanish↔English tasks, and obtain 24.1/30.3 and 28.1/27.6 BLEU points separately, outperforming state-of-the-art unsupervised results by more than 5 BLEU points and reducing the gap between unsupervised translation and supervised translation up to 50%.

## 1 INTRODUCTION

Neural Machine Translation (NMT) is a great success of deep learning for natural language processing. Thanks to recently developed advanced neural network architectures (Cho et al., 2014; Sutskever et al., 2014; Bahdanau et al., 2015; Jean et al., 2015; Vaswani et al., 2017; Gehring et al., 2017), NMT has significantly outperformed statistical machine translation and reached near human-level performance for several language pairs (Wu et al., 2016; Hassan et al., 2018). Such breakthroughs heavily depend on the availability of large scale of bilingual sentence pairs. Taking the WMT14 English→French task as an example, NMT uses 38 million parallel sentence pairs for training (Vaswani et al., 2017). As bilingual sentence pairs are costly to collect, the success of NMT is not fully realized on the vast majority of language pairs, especially for low-resource languages. Recently, Artetxe et al. (2017); Lample et al. (2017) tackle this challenge by training NMT models using only monolingual data, which achieves considerably good accuracy but still far away from that of the state-of-the-art supervised models.

While it is costly to collect bilingual sentence pairs by human translation, we notice that there exist many weakly paired bilingual documents on the Web. For example, for the entity "machine learning", Wikipedia has multiple articles in different languages, e.g., the English article and German article. The two articles have very similar content, but they are not sentence-by-sentence translations, since they may be independently created by different people. Similarly, an English news in BBC and a Chinese news in China Daily talk about the same event but maybe with differences in details. Furthermore, a popular novel in different languages is usually liberal translation instead of literal translation. We call such weakly aligned documents *weakly paired bilingual documents*. In this paper, we explore a new direction of learning NMT models from weakly paired documents, which has several advantages. First, weakly paired documents are much easier to obtain than bilingual sentence pairs. We can obtain bilingual document pairs from Wikipedia pages, from aligned news articles on international news websites, even from books. Second, such weakly paired documents have great coverage of different languages. For example, Wikipedia covers 178 languages and most

of them have paired pages to English. This means that learning translation models from paired bilingual documents are possible for many language pairs.

Weakly aligned document pairs can be utilized for NMT training from two aspects. First, although such two documents are not exactly sentence-by-sentence translations, it is possible that one specific sentence in one document is the translation of one sentence in the other document. Such a sentence pair can be used as bilingual signals for model training. For example in Wikipedia, although the web structure and paragraphs are generally different for entity "Beijing" in English and "Pekin" in French, we found the first sentences of the pages are quite semantically similar, in which both of them define Beijing as the capital of China (*English: Beijing, formerly romanized as Peking, is the capital of the People's Republic of China. French: Pekin, galement appele Beijing, est la capitale de la Rpublique populaire de Chine.*) The challenge is how to mine such sentence pairs from those document pairs. Second, although the sentences in two weakly paired documents are not aligned, the topics of the documents are well aligned. Such topic alignment is a strong signal that can be used to train NMT models.

In this paper, we focus on Wikipedia data and propose a method to train the machine translation models by leveraging weakly paired bilingual documents from Wikipedia. The key idea of our method is to mine implicitly aligned sentence pairs and leverage topic alignment as regularization. First, we provide a simple and efficient method to mine bilingual sentence pairs from each weakly aligned document pair. We first train cross-lingual word embeddings from two monolingual corpora (one for each language) using MUSE (Conneau et al., 2017), and then use weighted average of embeddings of words in a sentence as the sentence embedding. With the sentence embedding, we select the sentence pairs with large cosine similarity as bilingual sentence pairs and use them as supervised signals to train NMT models. Second, many previous works suggest that the word distribution can be used to well characterize the topic of the document (Petterson et al., 2010; Du et al., 2015; Funatsu et al., 2014; Pedrosa et al., 2016; Chemudugunta et al., 2007). To leverage the topic consistency between two weakly paired documents, we minimize the KL-divergence of the word distributions between the ground-truth document and the model-generated document.

Taking Wikipedia corpus as the training data, we test our method on the widely used WMT16 German↔English and WMT13 Spanish↔English translation tasks. Our method achieves 24.1/30.3 BLEU points for WMT16 German↔English translations and 28.1/27.6 BLEU points for WMT13 Spanish↔English translations, outperforming the state-of-the-art unsupervised method by more than 5 BLEU points and reduce the gap between unsupervised translation and supervised translation up to 50%.

## 2 RELATED WORK

Using monolingual data to boost the machine translation performance has attracted a lot of attention in the literature (Gulcehre et al., 2015; Sennrich et al., 2016a; Zhang & Zong, 2016; Wu et al., 2018; He et al., 2016), especially when the bilingual supervision is limited. Sennrich et al. (2016a) propose the back-translation approach which is a popular and effective way to augment the training bilingual sentence pairs with the target-side monolingual data. He et al. (2016) leverage both the source-side and target-side monolingual data in a dual learning framework. However, these methods still require a relatively large amount of labeled bilingual data. Recently, Lample et al. (2017) and Artetxe et al. (2017) make an initial study of unsupervised machine translation, in which the model is trained from the monolingual data only. Lample et al. (2017) leverage two key components to learn translation models from monolingual data: 1) suitable initialization of the translation model by cross-lingual word embeddings, 2) denoising auto-encoder as language model and reconstruction loss based on translation-back-translation. Both works leverage monolingual sentences only but do not leverage rich weakly paired documents from Web.

We study how to leverage document pairs for learning translation without sentence pairs and implement the proposed approach using Wikipedia data. Leveraging the free online Wikipedia database as an additional source to improve the natural language processing tasks has also attracted interest in recent years. For example, Conneau et al. (2017) show that word translation can be effectively learned based on the embedding trained from Wikipedia. This embedding further becomes one of the key components for unsupervised machine translation. Different from using Wikipedia to train warm-start word embeddings, we aim to leverage more and stronger signals from such weakly paired

documents to train translation model without parallel sentence pairs. Besides, Hálek et al. (2011) use the category information in Wikipedia corpus to improve the translation of named entities. Drexler et al. (2014) incorporate language models from target language documents that are comparable to the source documents in Wikipedia pages to improve the document translation.

There are several works aim at extracting potential sentence pairs from comparable corpus, but most of them rely on a set of bilingual sentence pairs to train a model and use this model to select sentence pairs. For example, Adafre & De Rijke (2006) and Yasuda & Sumita (2008) use a strong machine translation system to obtain a rough translation of a given page in one language into another, and then calculate word overlap or BLEU score between sentences as measure. Smith et al. (2010) and Munteanu & Marcu (2005) develop a ranking model/binary classifier to learn how likely a sentence in target language is a translation of the source language using parallel corpora. However, in our scenario, we have no bilingual sentence pairs available. That is, we have no bilingual sentence pairs to train such a model to further select new sentence pairs.

Our work is also related to document-to-document translation (Tu et al., 2018), but with different goals and settings. The goal of document-to-document translation is to enhance sentence-to-sentence translation with stronger signals beyond sentence pairs by using richer inputs, e.g., the topic information from the document that contains this sentence. During training, it takes one sentence as well as the cross-sentence (document-level) information as input and predicts the ground-truth translation sentence in other languages. Therefore, training a document-to-document translation model requires bilingual sentence pairs and their surrounding contexts. In our scenario, our goal is to learn a sentence-to-sentence translation model with weaker signals than sentence pairs. We target to extract useful information from weakly paired documents to train a translation model without human-labeled bilingual data.

## 3 OUR METHOD

We develop two ways to leverage weakly paired documents: mining implicitly aligned sentence pairs from the document pairs and aligning the topic distributions of two documents in a weakly aligned pair. Before diving into details, we first introduce some notations.

Denote $D = \{(d_i^X, d_i^Y)\}$, $i \in \{1, 2, ..., M\}$ as the set of weakly paired documents, in which document $d_i^X$ is aligned to $d_i^Y$. For example, $d_i^X$ and $d_i^Y$ are two cross-lingual linked Wikipedia pages. Denote $n_i^X$ and $n_i^Y$ as the number of sentences in document $d_i^X$ and $d_i^Y$ respectively. Note that usually $n_i^X \neq n_i^Y$.

Without any confusions, we denote $x$ as a sentence in language $X$ and $y$ as a sentence in language $Y$. We denote *enc* as the encoder for language $X$ and $Y$, which maps a sentence $x$ or $y$ into a sequence of real vectors using parameter $\theta_{enc}$. We use *dec* with parameter $\theta_{dec}$ as the decoder, which takes the encoded vectors and target language tag ($X$ or $Y$) as inputs and outputs a probability distribution over sentences in the target language. Let $\theta$ denote all the parameters of the translation model. Similar to Artetxe et al. (2017); Lample et al. (2017), such a model can handle both $X \rightarrow Y$ translation and $Y \rightarrow X$ translation.

### 3.1 MINING IMPLICITLY ALIGNED SENTENCE PAIRS

Different language versions of Wikipedia pages about the same entity/event are usually created by different people speaking different native languages, and therefore most sentences in two weakly aligned documents are not aligned. Even though, there is still a small chance that some bilingual sentences are aligned, and we try to mine such implicitly aligned sentence pairs and use them as supervision for NMT model training.

Imankulova et al. (2017) extract bilingual sentence pairs from Wikipedia using a well-trained translation model learned from supervised sentence pairs. This method does not work for us since we do not have aligned bilingual sentence pairs. Instead, our idea is to compute the similarity of two bilingual sentences using their cross-lingual sentence embeddings and choose the pairs with large similarity as aligned bilingual sentence pairs.

Sentence embedding is widely used to measure textual similarity in text classification tasks (Arora et al., 2017; Le & Mikolov, 2014; Wieting et al., 2015). Arora et al. (2017) compute the weighted average of the word embedding in one sentence where the weight depends on word frequency and then project away the weighted average sentence embeddings from their first principal component. This method achieves good performance on a range of monolingual textual similarity task. We extend their method from monolingual sentence embedding to cross-lingual sentence embedding, given the cross-lingual word emebddings are pre-trained using MUSE (Conneau et al., 2017). The detailed method is described as follows.

For each word $w$, we denote $e_w$ as the word embedding trained from MUSE (Conneau et al., 2017), $p(w)$ as the estimated frequency from another document and $a$ as a predefined parameter to calculate the weight of word embedding. We denote $\hat{e}_s$ as the weighted average sentence embedding of sentence $s$ and $E$ as the embedding matrix of all the sentences over the monolingual corpus. Then we remove the first principal components $u_1$ of $E$ for every weighted average sentence embedding $\hat{e}_s$ and use the resulting embedding $e_s$ as the final sentence embedding, i.e.,

$$\hat{e}_s = \sum_{w \in s} \frac{a}{a + p(w)} e_w, \tag{1}$$

$$u_1 \rightarrow PCA(E), \tag{2}$$

$$e_s = \hat{e}_s - u_1 u_1^T \hat{e}_s. \tag{3}$$

Based on the sentence embedding, we estimate the similarity between two sentences in different languages by their cosine similarity $\text{sim}(s^X, s^Y) = \frac{<e_{s^X}, e_{s^Y}>}{\|e_{s^X}\| \|e_{s^Y}\|}$. For each weakly aligned document pair $(d_i^X, d_i^Y)$, we have $n_i^X \times n_i^Y$ pairs of sentences and form a bipartite graph between sentences in two documents where the weight of an edge between two cross-lingual sentences is their cosine similarity score. The goal is to find the most confident edges (sentence pairs) from this weighted bipartite graph. We adopt a greedy selection approach with two constraints: The first constraint is that the weight of a selected edge must be larger than threshold $c_1$, which is to ensure that the two sentences are similar enough. The second constraint is that the weight of a selected edge must be larger than the weights of all other edges connected to these two nodes (sentences) by threshold $c_2$. This ensures that the pair we selected is unique enough. Denote $S = \{(s_j^X, s_j^Y)\}$ as the set of selected sentence pairs. We use those pairs as supervision for model training, i.e., minimizing the negative log-likelihood as below.

$$L_p(S; \theta) = \frac{1}{|S|} \sum_{(s^X, s^Y) \in S} \log P_{X \rightarrow Y}(s^Y | s^X; \theta) + \frac{1}{|S|} \sum_{(s^X, s^Y) \in S} \log P_{Y \rightarrow X}(s^X | s^Y; \theta) \tag{4}$$

## 3.2 Aligning Topic Distribution

Although cross-lingual linked Wikipedia pages are not aligned in sentences, they are usually aligned in topic distribution because they talk about the same event or entity. For example, the English topical words "politician", "United State" and "president" will appear in the English page for "Donald Trump", and similar topical words in French will appear in the corresponding French page. That being said, if we translate an article from English to French sentence-by-sentence, the word distribution of the translated article should be generally similar to the word distribution of the corresponding article in French. Here we leverage the document-level word-distribution alignment to enhance and regularize the training of a translation model.

Given an NMT model, we first translate a document $d_i^X$ through sentence translation and obtain a document $\bar{d}_i^Y$. Then we evaluate the word distributions between the generated document $\bar{d}_i^Y$ and the ground-truth document $d_i^Y$, and use such signal to optimize the model. However, straight-forward loss design, e.g., KL-divergence or Wasserstein distance between the word distributions of $\bar{d}_i^Y$ and $d_i^Y$ is not differentiable with respect to the NMT model, due to the non-differentiable operation (greedy search or beam search) while generating $\bar{d}_i^Y$.

To address this challenge, we need to design some loss function that is smooth with respect to the model parameters. Our proposal is assuming each generated sentence $\hat{s}_{i,k}^Y \in \bar{d}_i^Y$ is fixed and "refeeding" the pair $(s_{i,k}^X, \hat{s}_{i,k}^Y)$ to the model to get the probability distribution over all the words at each position. We calculate the word distribution by averaging word probability distributions over

all sentences and positions of the generated document. Mathematically, we have

$$P(w_{i,k,t}^Y | s_{i,k}^X, \hat{s}_{i,k,<t}^Y) \quad \sim \quad P_{X \to Y}(w_t^Y | s_{i,k}^X, \hat{s}_{i,k,<t}^Y; \theta), \tag{5}$$

$$P(w^Y; d_i^X, \theta) \quad \propto \quad \sum_{i,k,t} P(w_t | \hat{s}_{i,k,<t}^Y), \tag{6}$$

where $P(w^Y; d_i^X, \theta)$ "acts" as the model-generated word distribution for document $d_i^X$. Note that this averaged distribution is differentiable to model parameters. The word distribution of the ground-truth document $d_i^Y$ is $P(w^Y; d_i^Y) = \frac{\#w \text{ in } d_i^Y}{\#token \text{ in } d_i^Y}$. We simply use KL-divergence loss as the objective. Then the document alignment loss for $X \to Y$ translation is defined as

$$L_d(D; \theta, X \to Y) = \frac{1}{|D|} \sum_{(d_i^X, d_i^Y) \in D} KL(P(w^Y; d_i^Y) || P(w^Y; d_i^X, \theta)). \tag{7}$$

The corresponding document loss for $Y \to X$ translation can be defined in the same way. The final loss is as follows, which can be optimized using backpropagation thanks to its smoothness.

$$L_d(D; \theta) = L_d(D; \theta, X \to Y) + L_d(D; \theta, Y \to X) \tag{8}$$

### 3.3 OVERALL ALGORITHM

In addition to the above two ways of using Wikipedia data, the sentences in the weakly paired documents can be used as monolingual data to optimize the losses of the unsupervised machine translation. Therefore, our proposed losses can be combined with the loss functions of unsupervised machine translation. Here we first recap unsupervised machine translation (Lample et al., 2017), and then present our overall algorithm in Algorithm 1.

The unsupervised machine translation considers two loss functions. Given a monolingual sentence $s$ in language $X/Y$, the denoising auto-encoder loss is defined as $L_{dae} = \mathbb{E}_{s \sim X}[\log P(s|c(s); \theta)] + \mathbb{E}_{s \sim Y}[\log P(s|c(s); \theta)]$, where $c(.)$ is to drop and swap words in sentence $s$. As for the reconstruction loss, given $s$ in language $X/Y$ and the translated sentence $s'$ in $Y/X$ by model $P_{X \to Y}/P_{Y \to X}$, the reconstruction loss is defined as $L_{rec} = \mathbb{E}_{s \sim X}[\log P_{Y \to X}(s|s'; \theta)] + \mathbb{E}_{s \sim Y}[\log P_{X \to Y}(s|s'; \theta)]$.

Denote the combination of the monolingual data in language $X$ and $Y$ as $M$. We define the overall loss on monolingual data $M$ as $L_m(M; \theta) = L_{dae} + L_{rec}$. Finally, the overall training objective of our algorithm is to minimize the following loss function with hyperparameters $\alpha$ and $\beta$:

$$L = L_m(M; \theta) + \alpha L_p(S; \theta) + \beta L_d(D; \theta). \tag{9}$$

---

**Algorithm 1** Training Algorithm

---

**Require:** Initial translation model with parameter $\theta$; monolingual dataset $M$, implicitly aligned sentence pairs dataset $S$, weakly paired documents dataset $D$; optimizer *Opt*
1: **while** not converged **do**
2:     Randomly sample a mini-batch monolingual sentences from $M$, implicitly aligned sentence pairs from $S$ and weakly paired documents from $D$
3:     Calculate loss $L_m$, $L_p$ and $L_d$
4:     Update $\theta$ by minimizing the objective Eqn. (9) using optimizer *Opt*
5: **end while**

---

## 4 EXPERIMENTS

We test our method on several benchmark translation tasks. We first describe the data preparation and experimental design, and then present the main results, followed by some deep studies.

### 4.1 DATA PREPARATION

Wikimedia offers free copies of all available contents on Wikipedia for multiple languages. We download the language specific Wikipedia contents[1] in XML format, and use WikiExtractor[2] to extract and clean the texts. The numbers of Wikipedia documents are listed in Table 1. We then use the sentence tokenizer from toolkit NLTK to generate segmented sentences from Wikipedia documents.

Many Wikipedia pages contain external links to the pages that describe the same entity but in different languages. We extract weakly paired documents using these external links. We filter out a document pair if any document in the pair contains less than 5 sentences. We also remove the sentences longer than 100 words. We conduct experiments on two language pairs: English-German (En-De for short) translation and English-Spanish (En-Es for short) translation. Statistics of the processed Wikipedia documents are provided in Table 1.

We use the monolingual data as in Lample et al. (2017; 2018) together with Wikipedia document pairs to train NMT models. For the En-De task, we use all available sentences from the WMT monolingual News Crawl datasets from year 2007 to 2017 containting about 50 million sentences for each language. For En-Es, we use News Crawl datasets from year 2007 to 2012 containing about 10 million sentences. The trainslation models are evaluated on *newstest 2016* dataset for En-De and *newstest 2013* dataset for En-Es which are widely used (Koehn & Knowles, 2017).

| Language | #Wiki Documents |
|---|---|
| English | 5,684,240 |
| German | 2,201,782 |
| Spanish | 1,389,469 |

| Task | #Document Pairs |
|---|---|
| English-German | 948,631 |
| English-Spanish | 836,564 |

Table 1: Statistics of Wikipedia data, including numbers of documents and weakly paired documents.

### 4.2 EXPERIMENTAL DESIGN

To mine implicitly aligned sentences from weakly paired documents, we use the open-sourced word embeddings trained by Fasttext (Joulin et al., 2016) and use MUSE[3] to build cross-lingual word embeddings. We then generate sentence embeddings with the weighted average of cross-lingual word embeddings and further remove the top-1 principal components of sentence embedding matrix as introduced in Section 3.1. We set the two thresholds $c_1 = 0.7$ and $c_2 = 0.1$ respectively when selecting sentence pairs, and the parameter $a$ to calculate the weight of word embedding is 0.001. To translate a document $d_i^X$ for topic alignment, we use greedy search.

For the training of translation models, the monolingual datasets, Wikipedia document pairs and mined sentence pairs are jointly processed by BPE (Sennrich et al., 2016b) with $60,000$ codes. For model initialization, we follow Lample et al. (2018), which uses cross-lingual BPE embeddings to initialize the shared lookup tables, and the cross-lingual BPE embeddings are trained by fastText with embedding dimension 512, a context windows of size 5 and 10 negative samples. We adopt the Transformer (Vaswani et al., 2017) architecture in our experiments. We stack 6 layers in both the encoder and the decoder. Following Lample et al. (2018), we share the lookup tables between the encoder and the encoder, and between the source and target languages. The dimension of the hidden state is 512. The weights $\alpha$ and $\beta$ of the loss functions are set to be 1 and 0.05. For training, we use Adam optimizer (Kingma & Ba, 2014) and the same learning rate scheduler as used in Vaswani et al. (2017). For decoding, we use beam search with beam width 4 and length penalty 0.6, and the BLEU (Papineni et al., 2002) score is measured by multi-bleu.perl script[4].

---

[1] For example, we download English/German Wikipedia contents from `https://dumps.wikimedia.org/enwiki`, and `https://dumps.wikimedia.org/dewiki/`.
[2] `https://github.com/attardi/wikiextractor`
[3] `https://github.com/facebookresearch/MUSE`
[4] `https://github.com/moses-smt/mosesdecoder/blob/master/scripts/generic/multi-bleu.perl`

| Method | En→De | De→En | En→Es | Es→En |
|---|---|---|---|---|
| Lample et al. (2017) | 9.6 | 13.3 | - | - |
| Yang et al. (2018) | 10.9 | 14.6 | - | - |
| NMT (Lample et al., 2018) | 17.2 | 21.0 | 19.7 | 20.0 |
| PBSMT (Lample et al., 2018) | 17.9 | 22.9 | - | - |
| PBSMT + NMT (Lample et al., 2018) | 20.2 | 25.2 | - | - |
| NMT + First Wiki Sentence | 16.3 | 19.3 | 17.3 | 18.3 |
| NMT + Document Translation | 12.0 | 14.9 | 14.5 | 15.3 |
| Ours | **24.2** | **30.3** | **28.1** | **27.6** |
| Supervised | 33.6 | 38.2 | 33.2 | 32.9 |

Table 2: BLEU scores compared with previous approaches. 'First Wiki Sentence' and 'Document Translation' settings are introduced in Section 4.3.

## 4.3 MAIN RESULTS

Our method is compared with several previous works (Lample et al., 2018; 2017; Yang et al., 2018) in Table 2. We also consider some simple and heuristic ways of using the weakly paired documents from Wikipedia as baselines. One baseline, referred as "NMT + First Wikipedia Sentence", is to directly use the first sentence of the aligned documents as an aligned sentence pair to train NMT model. The motivation behind it is that usually the first sentence of a Wikipedia document summarizes the main content of the document, which is more likely to be similar across languages. The second baseline, referred as "NMT + Document Translation", is to treat the weakly aligned documents as two long sentences and use them as a bilingual sentence pair to train NMT models.

From Table 2, our approach achieves BLEU score of 24.1 and 30.3 on En→De and De→En translations respectively, which are the state-of-the-art numbers. Previous best performance is achieved by Lample et al. (2018): combining phrase-based approach and neural machine translation together, they achieve 20.2 and 25.2 BLEU scores on En→De and De→En translation. Our approach outperforms their method by more than 4 and 5 BLEU points on En→De and De→En respectively. For the En-Es, we achieve 28.1 and 27.6 BLEU scores on En→Es and Es→En respectively, with more than 8 and 7 points improvement over unsupervised Transformer baseline models. For the two heuristic baselines, we find that they hurt the performance of NMT models with more than 2 points decrease in terms of BLEU score. The results indicate that the careful utilization of Wikipedia data is important, which is also verified by the superior performance of our approach.

Furthermore, we report the supervised result in the last row of Table 2. The supervised setting is conducted on the full training set on WMT16 En-De translation with 4.5 million parallel sentences, and WMT13 En-Es translation with 3.8 million parallel sentences. As shown in the table, our approach takes a big step towards the supervised result, and reduce the gap between unsupervised translation and supervised translation up to 50%.

## 4.4 FURTHER STUDIES

In this subsection, we provide an ablation study to our method, check the performance variance with respect to the thresholds for mining implicit sentence pairs, and finally present several cases of mined sentence pairs. The studies are conducted on the En-De translation pair.

**Ablation Study**

Our method leverages weakly paired documents in two ways. To better understand the importance of the two ways, we report results from an ablation study in Table 3. From the table, we can see that removing the topic alignment, the accuracy drops with more than 1 BLEU points. Without the implicitly aligned

| Our Method | En→De | De→En |
|---|---|---|
| with $L_p$ and $L_d$ | 24.2 | 30.3 |
| without $L_d$ | 22.9 | 28.7 |
| without $L_p$ | 18.5 | 23.3 |

Table 3: Ablation study of our method on English-German translation tasks.

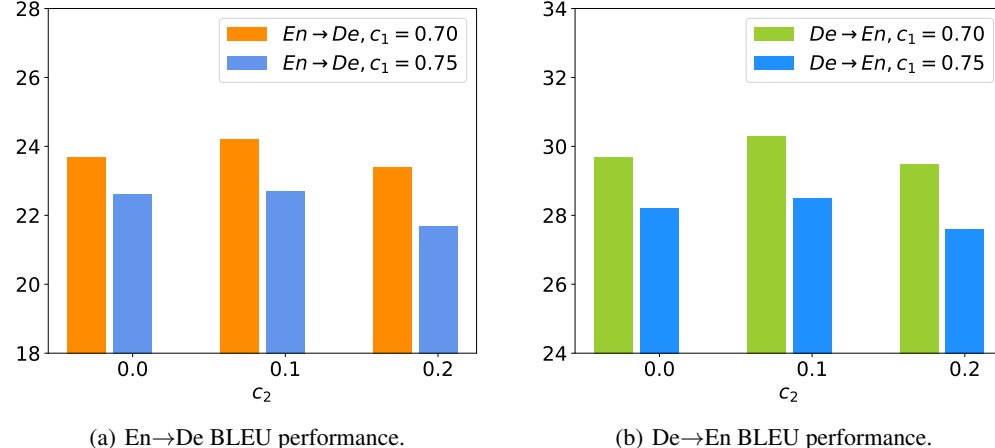

(a) En→De BLEU performance.     (b) De→En BLEU performance.

Figure 1: BLEU performance w.r.t. different thresholds for English-German translation pair.

sentence pairs, the accuracy decreases with about 6 BLEU points. These findings clearly demonstrate that both the two ways are important, and both contribute to the improvement of translation accuracy.

**Impact of Sentence Quality**

As introduced before, to control the data quality of our mined sentence pairs, we set two constraints with threshold $c_1$ and $c_2$. Here we present the paired sentence number and the final BLEU scores with respect to these two thresholds in Table 4 and Figure 1. We vary the threshold $c_1$ in $\{0.70, 0.75\}$ and $c_2$ in $\{0.0, 0.1, 0.2\}$. For $c_1 \leq 0.70$, we observe that the sentence quality is poor, while for $c_1 \geq 0.75$, we can only extract few pairs. As shown in the table and the figure, with more strict constraint, we obtain fewer sentence pairs but with higher quality. We can see that there is a clear trade-off between the data quality and the data number. A good configuration is $c_1 = 0.7$ and $c_2 = 0.1$.

| $\mathbf{c_1}/\mathbf{c_2}$ | English-German | | |
|---|---|---|---|
| | 0.0 | 0.1 | 0.2 |
| 0.70 | 257,947 | 199,965 | 132,403 |
| 0.75 | 100,497 | 84,271 | 58,814 |

Table 4: Number of the mined sentence pairs w.r.t. different thresholds for English-German.

**Case Study of Mined Sentence Pairs**

We present three cases of our selected sentence pairs in Table 5. We can see that our approach can mine high-quality pairs, such as the first case, in which one sentence is a good translation of the other one. Besides, our method can select interesting paired sentences with similar, if not the exactly same, semantics. As shown in the second case, the two sentences are almost semantically the same, while the German word *"gewählt"* (*"elected"* in English) is not included in the corresponding English sentence. Also in the last case, the detailed description *"15 Meter hoch"* (which means "15 meters high" in English) in the German sentence is missed in the English sentence. Although they are not exact translations, such sentence pairs are still very helpful for NMT model training, as demonstrated by the results of our method.

## 5 CONCLUSION AND FUTURE WORK

In this work, we proposed a general method to train neural machine translation models using weakly paired bilingual documents from the Web, e.g., Wikipedia. Our approach contains two key components: mining the implicitly aligned sentence pairs and aligning topic distributions. Experiments on public test benchmarks verify the effectiveness of our method.

| English | There are several different types of roles to be used in different situations . |
|---------|--------------------------------------------------------------------------------|
| German | Es gibt verschiedene Arten von Rollen , die in unterschiedlichen Situationen eingesetzt werden . |
| English | He was a member of the Swedish Academy from 1912 . |
| German | 1912 wurde er zum Mitglied der Schwedischen Akademie **gewählt** . |
| English | The statue and its marble base stand tall . |
| German | Die Statue ist , mit ihrem Sockel aus Marmor , **15 Meter hoch** . |

Table 5: Cases-studies of the sentence pairs mined by our approach for English-German.

For future work, we will apply our method to more language pairs, such as other hundreds of languages supported by Wikipedia. Second, we will study unsupervised machine translation using weakly paired documents from other data resources, such as news websites. Third, we will investigate better ways to utilize such weakly paired documents, going beyond mining sentence pairs and aligning topic distributions.

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
