# OpenReview forum: "Machine Translation With Weakly Paired Bilingual Documents"
_ICLR.cc/2019/Conference_

### Official Review · AnonReviewer2 · 2018-11-01
**nice contribution**

**Rating:** 7
**Confidence:** 5

**Review:**

Summary
The authors propose a relatively simple approach to mine noisy parallel sentences which are useful to greatly improve performance of purely unsupervised MT algorithms.
The method consists of a) mining documents that refer to the same topic, b) extracting from these documents parallel sentences, c) training the usual unsup MT pipeline with two additional losses, one that encourages good translation of the extracted parallel sentences and another one forcing the distribution of words to match at the document level.

Novelty: the approach is novel.

Clarity: the paper is clearly written.

Empirical validation: The empirical validation is solid but limited. The authors could further strengthen it by testing on low-resource language pairs (En-Ro, En-Ur).
It would also be useful to report more stats about the retrieved sentences in tab. 1 (average length compared to ground truth, BLEU using as reference the translation of a SoA supervised MT method, etc.)

Questions
1) Sec. 3.2 is the least clear of the paper. The notation of eq. 7 is quite unclear because of the overloading (e.g., P refers to both the model and the empirical distribution).
I am also unclear about this constraint about matching the topic distribution: as far as I understood, the model gets only one gradient signal for the whole document. I find then surprising that the authors managed to get any significant improvement by adding this term.
Related to this term, how is it computed? Are documents translated on the fly as training proceeds? Could the authors provide more details?

2) Have the authors considered matching sentences to any other sentence in the monolingual corpus as opposed to sentences in the comparable document?

---

> ### Author Response · Authors · 2018-11-21
> **Rebuttal from Authors**
>
> Thanks for your reviews and comments!
>
> 1. Regarding the empirical validation
> Thanks for the suggestions! We are conducting more experiments to low-resource language pairs. However, we want to make it clear that in machine translation, the low-resource tasks are usually referred to as learning a translation model with a small set of (or no) supervised bilingual sentence pairs [1,2,3]. From this perspective, the task, method, and experiment (En-De, De-En, En-Es, Es-En) in our paper are focused on low-resource problems indeed, as we use no human-labeled bilingual sentence pairs but only use document-level information or automatically mined related sentence pairs (maybe not exact or perfect translations).
> Our method shows that given no bilingual translation pairs but weakly paired documents (from Wiki, news websites or books), we can learn a translation model which is much better than the state-of-the-art unsupervised translation methods. As you suggested, we are currently working on the En-Ro and En-Tr language pairs, but since the document aligning process is costly, we are afraid that we may not give a result by the end of the rebuttal phase. We will report the number once the experiments are finished.
>
> 2. On the quality of retrieved sentences
> Actually, we have no *ground truth* for the retrieved sentence pairs, so we just use a well-trained translation model from huge bilingual data and check whether the retrieved sentences are *similar* to the translated sentences in terms of BLEU score in the below table.
> 	     En-De (c1=0.7)	   De-En (c1=0.7)
> C2=0.0	  26.86	            28.33
> C2=0.1	  30.68	            32.10
> C2=0.2	  33.40	            34.38
>
> As we expected, the sentence pairs we mined with more strict thresholds are more *similar* to the supervised model outputs. Besides, the reasonable BLEU scores show that the sentence pairs we extracted will be good to use in NMT model training.
>
> 3. Regarding the notion of P
> Sorry to make you feel confused. P is generally used as a notation of **probability**, but not for any specific parametric function. For example, in Eqn. 7, the first item P(w^Y; d^Y_i) refers to the empirical distribution of word w in language Y, and the second item P(w^Y; d^X_i, \theta) refers to the distribution of word w translated from document X_i with parameter \theta. We will modify the related equations to make them clearer.
>
> 4. Regarding the topic distribution implementation
> Your understanding is correct. In our experiments, during training, we generate the translated documents online to compute the topic distribution loss over document pairs, and the gradient signal is computed over a mini batch of document pairs.
>
> 5. Regarding sentence pairs in pure monolingual data
> Yes. In fact, we have tried this before the submission as it is a natural way to generalize our method to wider settings. We have tried to select sentence pairs over 50M monolingual WMT En-De dataset. According to our manual check, the quality of the sentence pairs we mined from the original monolingual dataset is not good. That’s why we mine the sentences from the weakly paired documents in our work.
>
> [1] Universal Neural Machine Translation for Extremely Low Resource Languages， Jiatao Gu†∗ Hany Hassan‡ Jacob Devlin∗ Victor O.K. Li†NAACL 2018
> [2] Neural Machine Translation for Low Resource Languages using Bilingual Lexicon Induced from Comparable Corpora, Sree Harsha Ramesh and Krishna Prasad Sankaranarayanan, NAACL, workshop 2018
> [3] Neural machine translation for low-resource languages, Robert Ostling ¨ and Jorg Tiedemann. 2017.

---

> > ### Author Response · Authors · 2018-11-25
> > **More Results on More Language Pairs**
> >
> > Dear Reviewer 2:
> >
> > Due to time limitation, we just provide some preliminary experimental results on En-Ro task. The results also show that the translation quality of our proposed method is better than that of the baseline. As the model performance is still growing, we will keep training and report the number when the optimization converges.
> >
> > Conducting this experiment requires a relatively long time mainly due to the following reasons:
> > 1. The cleaned Wikipedia dump file does not contain the internal link between the pages in different languages. We need to crawl the Wikipedia page online, match the pages between different languages and map such relationship back to the cleaned parsed Wikipedia contexts (https://en.wikipedia.org/wiki/Wikipedia:Database_download). Such a process is time-consuming.
> > 2. According to our experience, training an unsupervised NMT baseline model as well as our model needs more than **two weeks** using 4 GPUs to get a reasonable number.
> > Note that step 1 and 2 have dependencies.
> >
> > To address the concerns from the reviewers on the adaptability of our proposed method, we quickly conducted experiments with **a small number of paired documents** (tens of thousands), **a small number of mined sentences** (several thousands) extracted from step 1. Then we train the model using such data as well as monolingual data for *one week* directly using the configuration of En-De/En-Es with no hyperparameter tuning.
> >
> > To be fair enough, we compare our method with the unsupervised baseline with the same training time. For direction En->Ro, the BLEU score of the unsupervised baseline trained for one week achieves about 10.33 and we achieve 12.60 using our method. For direction Ro-En, we achieve BLEU score 15.98 while the baseline is 12.63.
> > We believe with more data in step 1 and longer training time in step 2, our proposed method will have more improvements. The current experimental results already show that our method has great potential and is robust to handle more language pairs.

---

> > > ### Comment · AnonReviewer2 · 2018-11-27
> > > **thank you for the response**
> > >
> > > I have read your response as well as the other comments. Thank you for taking the time to respond to my questions.
> > >
> > > I think the Authors should revise their paper to a) include the references mentioned by the other reviewers and AC and b) strengthen their empirical validation.
> > >
> > > I agree with the Authors that this contribution is different from past work because it is in the fully unsupervised setting. However, I also think that the AC made a good point in saying that for all language pairs for which there is good and abundant alignment between Wikipedia pages, there is also likely to be some parallel data, which diminishes the practical impact of this work.
> > > I also do not agree with the Authors definition of "low-resource" language pair. En-De is not low resource by any standard definition, as anybody can easily find abundant amount of parallel sentences. It is fine to simulate unsupervised MT in these high resource languages, but it is not fine to call these examples of low resource languages.  For instance, En-Ur has instead little parallel data in domains of interest (e.g., news). Proving your method on such languages is the ultimate test.
> > >
> > > Despite all of this, I still think that this work has merit and is good enough to be presented at the conference (assuming references are added and properly contrasted in the revision), as it can sparkle discussion and promote research in this space which may make this effort more practical.

---

### Official Review · AnonReviewer3 · 2018-11-02
**the claimed "new direction" has been explored before.**

**Rating:** 5
**Confidence:** 5

**Review:**

The major issue in this paper is that the "new direction" in this paper has been explored before [1]. Therefore the introduction needs to be rewritten with arguing the difference between existing methods.

The proposed method highly relies on the percentage of implicitly aligned data. I suggest the author do more experiments on different data set with a significant difference in this "percentage". Otherwise, we have no idea about the performance's sensitivity to the different datasets.

More detailed explanations are needed. For example, what do you mean by "p(w)  as the estimated frequency"? Why do we need to remove the first principal components?

Section 3.2 title is " aligning topic distribution" but actually it is doing word distribution alignment.

Do you do normalization for P(w^Y;d_i^X,\theta) in eq.6 which is defined on the entire vocab's distribution?

I think the measurement of the alignment accuracy and more experiments with different settings of \alpha and \beta are needed.

Citation needed for "Second, many previous works suggest  that the word distribution ..."

[1] Munteanu et al, "Improving Machine Translation Performance by Exploiting Non-Parallel Corpora", 2006

---

> ### Author Response · Authors · 2018-11-21
> **Rebuttal from Authors - Part 1**
>
> We thank Reviewer 3 for the reviews and comments! Here are our responses to the concerns.
>
> 1. Regarding the related work
> Thanks for the reference. We are indeed aware of the related work on selecting sentence pairs from monolingual corpora. We did try some methods and found them do not work well as the scenario of related works is far different from ours.
> The methods in [1-4] rely on bilingual sentences to train a model and use this model to select sentence pairs. For example, [1-2] use an MT system to obtain a rough translation of a given page in one language into another and then uses word overlap or BLEU score between sentences as measures.  [3-4] develop a ranking model/binary classifier to learn how likely a sentence in the target language is the translation of a sentence in the source language using parallel corpora. However, in our setting, we don’t have any bilingual sentences pair available. That is being said, we have no bilingual sentence pairs to train such a model to further select new data pairs.
> In order to work similarly to the previous works in the unsupervised setting, the most related model for selecting pairs is the unsupervised machine translation model. We did try to use an unsupervised translation model for sentence pair selection at the very early stage of the work. We first trained an unsupervised model followed [7] and then use the model outputs to evaluate each sentence pairs between two linked documents. We have conducted the following experiments:
> (a). Similarly to [1,2], for each sentence x, we generate the translation results using the unsupervised NMT model, select the most similar sentence to the translation results (in terms of BLEU), and use such data pairs for NMT training.
> (b). To build up a scoring function as used in [3-4], we use the model-output probability as the scoring function. We select sentence pair (x, y) with larger translation probabilities p(y|x) and use such data pairs for NMT training.
> As the **unsupervised translation model** is not good enough, the selected sentence pairs are not reasonable as shown in the below table. We hypothesize this is due to that as some sentences in one Wiki pages are similar (e.g., a few words differ from each other), then 1. the BLEU(or sentence-level BLEU) score is very sensitive to evaluate such sentences. 2. the likelihood on similar sentences are not that trustable.
> Furthermore, we found training an NMT model using such poor data does not work well.  On WMT De-En task, we have the following results: The BLEU score of model trained in (a) can only reach 22.4. The best model trained in (b) can achieve only 19.8 in terms of BLEU score. Both show that the trained NMT models are not good as expected.
> As a summary, we find by leveraging the recent techniques (the cross-lingual word embedding + unsupervised sentence representation), the selected sentences are much better. We believe our findings are important to the field of unsupervised learning and unsupervised machine translation. We will include those discussions in our paper and clarify the differences between our work and previous works.
>
> English	|| Selected German sentence by unsupervised translation model	|| Selected German sentence by our method
> She was one of the pioneers of Greek surrealism .	|| Inzwischen ist sie Mitglied der Kommunistischen Partei geworden .	|| Zunächst zählt sie zu den Pionieren des griechischen Surrealismus .
> The film premiered at the 2014 Zurich Film Festival .	|| In Deutschland startete der Film am 10. September 2015 .	|| Er hatte seine Premiere am 26. September 2014 beim Zurich Film Festival .
> The eastern part is leafy and park-like .	|| Außerdem befindet sich hier ein Kinderspielplatz .	|| Der östliche Teil ist begrünt und parkähnlich gestaltet .
> Most of the remaining convicts were then relocated to Port Arthur .	|| Insgesamt wurden in der Strafkolonie 1200 Häftlinge verwahrt .	|| Die verbliebenen Häftlinge wurden schrittweise ins Lager nach Port Arthur verlegt .
>
> 2. Regarding more experiments on the different percentage of data pairs
> We are afraid that you might miss some parts of our paper. We have tested the performance with different percentage of implicitly aligned data according to the different choices of the thresholds to understand the sensitivity of the data size in Section 4.4. As we can see from Section 4.4, by setting different thresholds, we select data pairs from 60k to 250k. We think these results answer the question you mentioned. The different sizes of data indeed have impacts to the model performance, but all experimental results show that our model is better than the baselines (i.e., comparing the numbers in Figure 1 and the baselines in Table 2).

---

> ### Author Response · Authors · 2018-11-21
> **Rebuttal from Authors - Part 2**
>
> 3. Regarding why to remove the first principal component of the embedding and p(w)
> We follow the unsupervised sentence representation approach from [5,6] to remove the first principal component of sentence embedding but not word embedding, as mentioned in the 4th paragraph of Section 3.1. Intuitively, from empirical observations, the embeddings of many sentences share a large **common vector**. Removing the first principal component from the sentence embeddings make them more diverse and expressive in the embedding space, and thus the resulted embeddings are shown to be more effective [5,6].
> p(w) is the unigram probability (in the entire corpus) of the word w. We will make it clearer.
>
> 4. Regarding the notion of topic distribution, normalization, and citations
> We have added citations about the word distribution in the new paper version. We are just trying to describe that the topics between the source document and target document should be similar if they talk about the same event, and thus they should use similar words. We can change the term **topic distribution** to **word distribution** if you think it is essential and important.
> Apparently, we did the normalization over the target vocabulary as we use KL-divergence loss function.
>
> 5. More experiments on alpha and beta
> We made more analysis on the model trained with different alpha and beta on En-De data, and listed the numbers in the below table. We found that the value of alpha is robust to the model performance.
> Alpha	0.5	0.8	1.0	1.2	1.5
> En-De	23.6	24.0	24.2	24.1	24.1
> De-En	29.8	30.1	30.3	30.2	30.1
> We found larger values for beta will make model worse if the KL-divergence contributes much in the loss function. beta=0.05 is the best configuration we have found.
>
> [1] Adafre S F, De Rijke M. Finding similar sentences across multiple languages in Wikipedia[C]//Proceedings of the Workshop on NEW TEXT Wikis and blogs and other dynamic text sources. 2006.
> [2] Yasuda K, Sumita E. Method for building sentence-aligned corpus from wikipedia[C]//2008 AAAI Workshop on Wikipedia and Artificial Intelligence (WikiAI08). 2008: 263-268.
> [3] Smith J R, Quirk C, Toutanova K. Extracting parallel sentences from comparable corpora using document-level alignment[C]//Human Language Technologies: The 2010 Annual Conference of the North American Chapter of the Association for Computational Linguistics. Association for Computational Linguistics, 2010: 403-411.
> [4] Munteanu, Dragos Stefan, and Daniel Marcu. "Improving machine translation performance by exploiting non-parallel corpora." Computational Linguistics 31.4 (2005): 477-504. ", 2006
> [5] Arora, Sanjeev, Yingyu Liang, and Tengyu Ma. "A simple but tough-to-beat baseline for sentence embeddings." ICLR-2017.
> [6] Mu, Jiaqi, Suma Bhat, and Pramod Viswanath. "All-but-the-top: Simple and effective postprocessing for word representations." ICLR-2018.
> [7] Phrase-Based & Neural Unsupervised Machine Translation. EMNLP-2018.

---

### Official Review · AnonReviewer1 · 2018-11-04
**Nice BLEU score improvements over existing work but will it generalise to low-resource language pairs?**

**Rating:** 6
**Confidence:** 3

**Review:**

This paper proposes a method to train a machine translation system using weakly paired bilingual documents from Wikipedia. A pair of sentences from a weak document pair are used as training data if their cosine similarity exceeds c1, and the similarity between this sentence pair is c2 greater than any other pair in the documents, under sentence representations formed from word embeddings trained with MUSE. The neural translation model learns to translate from language X to Y, and from Y to X using the same encoder and decoder parameters, but the decoder is aware of the intended target language given an embedding of the intended language. The model is also trained to minimise the KL divergence between the distribution of terms in the target language document and the distribution of terms in the current model output. The model also uses the denoising autoencoding and reconstruction objectives of Lample et al. (2017). The results show improvements over the Lample et al. (2017) and that performance is heavily dependent on the number of sentences extracted from the weakly aligned documents.

Positives
- Large improvement over previous attempts at unsupervised MT for the En-De language pair.
- Informative ablation study in Section 4.4 of the relative contribution of each part of the overall objective function (Eq 9).

Negatives
- The introduction gave the impression that this method would be applied to low-resource language pairs but it was applied to two high-resource language pairs. Because you have not evaluated on a low-resource language pair, it's not clear how your proposed method would generalise to a low-resource setting.

Questions
- Can you give some intuition for why you remove the first principal component from the word embeddings in Equations 1 - 3?
- Are the Supervised results in Table 2 actually a fair reflection of a reasonable NMT model trained with sub-word representations and back translated data?
- What is the total number of sentences in the weakly paired documents in Table 1? It would be useful to know the proportion of sentences you managed to extract to train your models.

Comments
- Koehn et al. (2003) is not an example of any kind of neural network architecture.

---

> ### Author Response · Authors · 2018-11-21
> **Rebuttal from Authors**
>
> We thank Reviewer 1 for the reviews and comments! Here are our responses to the concerns.
>
> 1. Regarding low-resource tasks
> We want to make it clear that in machine translation, the low-resource tasks are usually referred to as learning a translation model with a small set of (or no) supervised bilingual sentence pairs [1,2,3]. From this perspective, the task, method, and experiment (En-De, De-En, En-Es, Es-En) in our paper are focused on low-resource problems indeed, as we use no human-labeled bilingual sentence pairs but just use document-level information or automatically mined related sentence pairs (which may be not exact or perfect translations).
> Our method shows that given no bilingual translation pairs but weakly paired documents (e.g., from Wiki), we can learn a translation model which is much better than the state-of-the-art unsupervised translation methods.
>
> 2. Regarding why to remove the first principal component of the embedding
> We follow the unsupervised sentence representation approach from [4,5] to remove the first principal component of sentence embedding but not word embedding, as mentioned in 4th paragraph of Section 3.1. Intuitively, from empirical observations, the embeddings of many sentences share a large **common vector**. Removing the first principal component from the sentence embeddings make them more diverse and expressive in the embedding space, and thus the resulted embeddings are shown to be more effective [4,5].
>
> 3. Regarding the supervised baseline
> All the supervised models are trained on the widely acknowledged WMT bilingual dataset using Transformer [6], which is considered to be a standard baseline model of NMT tasks [7]. For our learned models and the baseline models, we do follow the common practice and use sub-word tokens (Byte Pair Encoding (BPE) approach) as in [6]. We have mentioned this in 2nd paragraph of Section 4.2.
>
> 4. Regarding data statistics
> For the number of sentences in the weakly paired documents, there are 4,285,607 English sentences and 4,266,178 German sentences in English-German language pair, 2,679,278 English sentences and 2,547,358 Spanish sentences in English-Spanish language pair. Therefore, we extract a reasonable proportion of sentences from the weakly paired documents to train the model.
>
> [1] Universal Neural Machine Translation for Extremely Low Resource Languages， Jiatao Gu†∗ Hany Hassan‡ Jacob Devlin∗ Victor O.K. Li†NAACL 2018
> [2] Neural Machine Translation for Low Resource Languages using Bilingual Lexicon Induced from Comparable Corpora, Sree Harsha Ramesh and Krishna Prasad Sankaranarayanan, NAACL, workshop 2018
> [3] Neural machine translation for low-resource languages, Robert Ostling ¨ and Jorg Tiedemann, 2017.
> [4] Arora, Sanjeev, Yingyu Liang, and Tengyu Ma. "A simple but tough-to-beat baseline for sentence embeddings." ICLR-2017.
> [5] Mu, Jiaqi, Suma Bhat, and Pramod Viswanath. "All-but-the-top: Simple and effective postprocessing for word representations." ICLR-2018.
> [6] Vaswani, Ashish, et al. "Attention is all you need." NIPS-2017.
> [7] Phrase-Based & Neural Unsupervised Machine Translation. EMNLP-2018.

---

> > ### Author Response · Authors · 2018-11-25
> > **More Results on More Language Pairs**
> >
> > Dear Reviewer 1,
> > Due to time limitation, we just provide some preliminary experimental results on En-Ro task. The results also show that the translation quality of our proposed method is better than that of the baseline. As the model performance is still growing, we will keep training and report the number when the optimization converges.
> >
> > Conducting this experiment requires a relatively long time mainly due to the following reasons:
> > 1. The cleaned Wikipedia dump file does not contain the internal link between the pages in different languages. We need to crawl the Wikipedia page online, match the pages between different languages and map such relationship back to the cleaned parsed Wikipedia contexts (https://en.wikipedia.org/wiki/Wikipedia:Database_download). Such a process is time-consuming.
> > 2. According to our experience, training an unsupervised NMT baseline model as well as our model needs more than **two weeks** using 4 GPUs to get a reasonable number.
> > Note that step 1 and 2 have dependencies.
> >
> > To address the concerns from the reviewers on the adaptability of our proposed method, we quickly conducted experiments with **a small number of paired documents** (tens of thousands), **a small number of mined sentences** (several thousands) extracted from step 1. Then we train the model using such data as well as monolingual data for *one week* directly using the configuration of En-De/En-Es with no hyperparameter tuning.
> >
> > To be fair enough, we compare our method with the unsupervised baseline with the same training time. For direction En->Ro, the BLEU score of the unsupervised baseline trained for one week achieves about 10.33 and we achieve 12.60 using our method. For direction Ro-En, we achieve BLEU score 15.98 while the baseline is 12.63.
> > We believe with more data in step 1 and longer training time in step 2, our proposed method will have more improvements. The current experimental results already show that our method has great potential and is robust to handle more language pairs.

---

> > > ### Comment · AnonReviewer1 · 2018-11-26
> > > **Re: More Results on More Language Pairs**
> > >
> > > Thank you for providing some preliminary results on a more challenging language pair, and for addressing the questions in my review. The setup you explore in this new experiment is particularly interesting because you only have tens of thousands of weakly paired documents, as compared to millions of documents.

---

### Public Comment · (anonymous) · 2018-10-05
**On Topic Distribution Loss**

Are there only success cases in your ablation study with BLEU being hampered by removing the topic loss part of the objective? The filtering indicates that non-comparable articles pass off as weakly paired documents - which can often lead to a wrong signal. Do you have any cases (success/failure) indicating the same? I see alpha as 1 and beta as 0.05 balances this to some extent.

Could you detail the stats further to include a bit of word level stats on the pair of documents somehow to see if these are comparable articles or not?

I'm particularly curious about how this would scale to the low resource setting, where the noisy loss signals becomes more prominent. How likely am I to get similar results?

Thanks,

---

> ### Author Response · Authors · 2018-11-23
> **Response from Authors**
>
> Thanks for your comment and sorry for the late response.
>
> 1. Regarding the document data quality and beta (for document loss) impacts
> As we have no ground truth of the paired data quality on Wiki, we simply checked the quality of our extracted data according to a well-trained supervised translation model. In particular, as we have extracted a set of paired sentences, we checked the BLEU score by comparing the translation from a well-trained supervised model and our mined sentence pairs. The results are in the below table.
> 	      En-De(c1=0.7)	   De-En(c1=0.7)
> C2=0.0	   26.86	                  28.33
> C2=0.1	   30.68	                  32.10
> C2=0.2	   33.40	                  34.38
> As you can see, the quality of our extracted sentence pairs is good and the reasonable BLEU scores show that the data will be good to use in NMT model training. Since these sentence pairs are extracted from the weakly paired documents, therefore we think such paired documents can provide valuable information for the model training. For the document loss, we vary the value of beta in the experiments, and we found if we use a very large beta, the KL-divergence loss will contribute much and dominant other loss terms, the trained model in such setting is a little worse than setting beta = 0.05 as used in our experiments.
>
> 2. Regarding to the low-resource setting
> We are conducting more experiments on En-Ro and En-Tr to test our proposed methods. We will report the number once the experiments are finished. However, we still want to make it clear in machine translation, low-resource tasks are usually referred to as learning a translation model with a small set of (or no) supervised bilingual sentence pairs [1,2,3]. From this perspective, the task, method, and experiment in our paper are focused on low-resource problems indeed, as we use no human-labeled bilingual sentence pairs but only document-level information or automatically mined related sentence pairs (maybe not exact or perfect translations). We show that given no bilingual translation pairs but weakly paired documents (from Wiki), we can learn a translation model which is much better than the state-of-the-art unsupervised translation methods.
>
> [1] Universal Neural Machine Translation for Extremely Low Resource Languages， Jiatao Gu†∗ Hany Hassan‡ Jacob Devlin∗ Victor O.K. Li†NAACL 2018
> [2] Neural Machine Translation for Low Resource Languages using Bilingual Lexicon Induced from Comparable Corpora, Sree Harsha Ramesh and Krishna Prasad Sankaranarayanan, NAACL, workshop 2018
> [3] Neural machine translation for low-resource languages, Robert Ostling ¨ and Jorg Tiedemann. 2017.

---

### Comment · Area_Chair1 · 2018-11-06
**Please Clarify Theoretical and Empirical Advantages over Previous Work**

As noted by Reviewer 3, the extraction of parallel sentences from comparable corpora has been covered extensively in the previous literature. While the method presented here is unquestionably useful, there is only a single reference to a paper from 2017, despite the fact that similar methods have existed and been widely studied since at least 2006. In order for me to recommend the paper for acceptance, I would like to see a comparison, theoretical and empirical, to the prominent previous works in the field of parallel sentence mining from comparable corpora, starting with the method cited by Reviewer 3 and also covering more recent work.

---

> ### Author Response · Authors · 2018-11-21
> **Response from Authors**
>
> Dear area chair, thanks for your comment!
> First, as discussed in our response to Reviewer 3, existing works rely on bilingual sentence pairs to train a model for parallel sentence mining. Note that we focus on the unsupervised setting, where there is no bilingual sentence pair and so existing methods cannot be applied.
> Second, we also tried to use an unsupervised neural machine translation model to rank/select sentence pairs from comparable corpora as in [1-4]. As described in our response to Reviewer 3, according to extended experiments and case studies, we find that the data quality of the sentence pairs selected by such unsupervised model  is not that good and the final trained translation model is worse than ours.
> As a summary, we find by leveraging the recent techniques (the cross-lingual word embedding + unsupervised sentence representation), the selected sentences are much better. We believe our findings are important to the field of unsupervised learning and unsupervised machine translation.
>
> [1] Adafre S F, De Rijke M. Finding similar sentences across multiple languages in Wikipedia[C]//Proceedings of the Workshop on NEW TEXT Wikis and blogs and other dynamic text sources. 2006.
> [2] Yasuda K, Sumita E. Method for building sentence-aligned corpus from wikipedia[C]//2008 AAAI Workshop on Wikipedia and Artificial Intelligence (WikiAI08). 2008: 263-268.
> [3] Smith J R, Quirk C, Toutanova K. Extracting parallel sentences from comparable corpora using document-level alignment[C]//Human Language Technologies: The 2010 Annual Conference of the North American Chapter of the Association for Computational Linguistics. Association for Computational Linguistics, 2010: 403-411.
> [4] Munteanu, Dragos Stefan, and Daniel Marcu. "Improving machine translation performance by exploiting non-parallel corpora." Computational Linguistics 31.4 (2005): 477-504. ", 2006.

---

> > ### Comment · Area_Chair1 · 2018-11-26
> > **Lots of existing work does not rely on sentence pairs**
> >
> > Hello,
> >
> > Thank you for the follow-up comment. However, there are many works that do not rely on bilingual sentence pairs, rather using translation lexicons only. One example of such a method is below, but a quick search should be able to reveal others:
> >
> > http://www.aclweb.org/anthology/W04-3208
> >
> > These translation lexicons can be easily extracted from, for example, Wikipedia language links. Given that Wikipedia (or other similar document pairs) is a resource requirement for the proposed method, it seems that the use of much stronger baselines would be possible, and in fact necessary to stress the utility of the proposed method.
> >
> > As an auxiliary note, it's not clear to me how realistic it is to have a high-quality document collection such as Wikipedia, but not have *any* parallel data that can be used to train a classifier such as the one used by Munteanu and Marcu. The reason why this method is widely used is because it works well and is less reliant on heuristics such as the ones in the paper above. Are there any languages in the world where we have a significantly sized Wikipedia or similar document collection, but don't have any sentence aligned parallel data whatsoever, even from sources such as the Bible, which is translated into 2500 languages? If there are, that would be a convincing argument for the utility of this method.

---

### Author Response · Authors · 2018-11-26
**Response to Reviewers and Area Chair**

Dear Reviewers and Area Chair,

Thanks again for your great reviews and comments. According to your suggestions, we have made more discussions and experiments and updated the paper. We list the main points as below.

1. We have provided preliminary experimental results on additional language pairs according to the suggestions from Reviewer 1 and Reviewer 2. According to the current results, our method is better than the baselines and we will keep tracking the status.
2. We have discussed the difference between our method and existing methods on sentence mining to address the concerns from Reviewer 3 and Area Chair with empirical comparisons.
3. We have updated our paper and added more discussions about related works according to the suggestions from Reviewer 3.

We hope our responses can help address your concerns and questions.

---

### Author Response · Authors · 2018-11-28
**Response to Reviewer 2 and Area Chair**

Dear Reviewer 2 and Area Chair:

Thanks for the comments. We will definitely revise our paper to add more related work and comparisons. We also want to make discussions about the following points.

1. For the experimental setting
1). We fully agree that the method should be tested on the setting you mentioned, which has no parallel data at all. However, please note that in order to verify the performance of a translation model, we need some **ground truth** in-domain sentence pairs for evaluation (Bible is out-of-domain), e.g, the most standard and widely used WMT and IWSLT test data. However, once we have a **ground truth test set**, there always exists corresponding training data, which are bilingual sentences. We think that finding a setting that satisfies 1. A significant number of documents. 2. No-parallel sentences anywhere. 3. Can be professionally evaluated is almost impossible.
In order to fairly compare with previous work and study the effectiveness of the proposed method, we follow all previous works to use these WMT translation test data for evaluation. Note that the previous work [4] also use En-De, even En-Fr to test their unsupervised models.

2. Regarding the contribution
1). We notice that almost all concerns are around the sentence mining approach. However, we kindly point out that sentence mining is only a component of our proposed method. We also leverage information in weakly paired documents and the ablation study shows that our document loss can improve the model performance.
2). Our work and previous works on unsupervised translation use Wiki data (e.g., [3,4]), which is shown to be very effective. [3] uses Wiki data to learn cross-lingual embeddings (which is also used in our work) and [4] use the cross-lingual embedding as a warm start to initialize the parameters used in translation models. However, [3] aims at solving a simpler task while [4] simply uses the Wiki data trained embedding as *parameter initialization*. Our method can be considered as taking one step further compared to the most recent previous works by better leveraging documents to learn sentence-level translation models.

3. On recent experiments on sentence mining approaches
Sentence pair mining approaches rely on a bilingual dictionary either from other resources or learned from supervised bilingual data, such as [1]. [2] (the paper Area Chair mentioned) also stated that they used parallel corpora to initialize their EM lexical, and they found that initialization with “very-non-parallel corpora” performs terrible (you can check the figure 3 in [2]).
We followed AC’s suggestion to get bilingual dictionary based on titles from cross-lingual Wikipedia language links, and did some experiments. We evaluate the generated bilingual dictionary using tools from https://github.com/facebookresearch/MUSE, which is a benchmark task to measure the quality of a learned dictionary. The results are as follows:
Source -> Target	Dict from Wiki (Top 1 Accuracy)
        En->Es	                      0.177
        Es->En	                      0.185
        En->De	                      0.129
        De->En	                      0.149

As you can see, the Top 1 accuracy of the dictionary is very poor, which shows the quality of the dictionary is not good. This is reasonable because titles of Wikipedia pages are usually entities, while a useful dictionary is always beyond entities. Our approach is based on calculating similarities between sentence embeddings, but we can also test for **word pair mining** using the word embeddings and the accuracy is more than 70% for the tasks. Therefore, it is obvious that our used approach is much better than the title dictionary.

[1] Munteanu, Dragos Stefan, and Daniel Marcu. "Improving machine translation performance by exploiting non-parallel corpora." Computational Linguistics 31.4 (2005): 477-504. ", 2006
[2] Fung, Pascale, and Percy Cheung. "Mining very-non-parallel corpora: Parallel sentence and lexicon extraction via bootstrapping and e." EMNLP 2004.
[3] Alexis Conneau, Guillaume Lample, Marc’Aurelio Ranzato, Ludovic Denoyer, and Herve Jegou. Word translation without parallel data. ICLR 2017
[4] Guillaume Lample, Ludovic Denoyer, and Marc’Aurelio Ranzato. Unsupervised machine translation using monolingual corpora only. ICLR 2017

---

### Meta-Review · Area_Chair1 · 2018-12-13
**Method is likely useful, but paper needs to be re-framed in light of previous work.**

**Confidence:** 4
**Recommendation:** Reject

**Metareview:**

This paper proposes a new method to mine sentence from Wikipedia and use them to train an MT system, and also a topic-based loss function. In particular, the first contribution, which is the main aspect of the proposal is effective, outperforming methods for fully unsupervised learning.

The main concern with the proposed method, or at least it's description in the paper, is that it isn't framed appropriately with respect to previous work on mining parallel sentences from comparable corpora such as Wikipedia. Based on interaction in the reviews, I feel that things are now framed a bit better, and there are additional baselines, but still the explanation in the paper isn't framed with respect to this previous work, and also the baselines are not competitive, despite previous work reporting very nice results for these previous methods.

I feel like this could be a very nice paper at some point if it's re-written with the appropriate references to previous work, and experimental results where the baselines are done appropriately. Thus at this time I'm not recommending that the paper be accepted, but encourage the authors to re-submit a revised version in the future.